# Correspondence-Free Point Cloud Registration with SO(3)-Equivariant Implicit Shape Representations

**Minghan Zhu[1], Maani Ghaffari[2], Huei Peng[1]**
[1]Department of Mechanical Engineering, University of Michigan, United States
[2]Department of Naval Architecture and Marine Engineering, University of Michigan, United States
{minghanz, maanigj, hpeng}@umich.edu

**Abstract:** This paper proposes a correspondence-free method for point cloud rotational registration. We learn an embedding for each point cloud in a feature space that preserves the SO(3)-equivariance property, enabled by recent developments in equivariant neural networks. The proposed shape registration method achieves three major advantages through combining equivariant feature learning with implicit shape models. First, the necessity of data association is removed because of the permutation-invariant property in network architectures similar to PointNet. Second, the registration in feature space can be solved in closed-form using Horn's method due to the SO(3)-equivariance property. Third, the registration is robust to noise in the point cloud because of the joint training of registration and implicit shape reconstruction. The experimental results show superior performance compared with existing correspondence-free deep registration methods.

**Keywords:** Point cloud registration, implicit shape model, equivariant neural network, representation learning

## 1 Introduction

Point clouds are a major form of 3D information representation in perception with numerous applications today, including intelligent robots and self-driving cars. The registration of two different point clouds capturing the same underlying object or scene refers to estimating the relative transformation that aligns them. Point cloud registration enables information aggregation across several observations, which is essential for many tasks such as mapping, localization, shape reconstruction, and object tracking. Given perfect point-to-point pairings between the point-cloud-pair, the relative transformation can be solved in closed-form using Horn's method [1].

However, such a pairing is not available or is corrupted in real-world problems, posing a significant challenge for the point cloud registration task, namely data association. ICP matches the closest points in Euclidean space and iteratively updates the matching using the current pose estimation [2]. In the deep learning era, the strong feature learning capacity of neural networks enables better description and discrimination of points, improving their matching. Overall, these methods can be categorized as *correspondence-based* registration.

An alternative approach, called *correspondence-free* registration, is to completely circumvent the matching of points [3] by learning a global representation for the entire point cloud and estimating the transformation by aligning their representations. Similar to correspondence-based registration, correspondence-free registration can also be with or without the involvement of deep learning. The main obstacle in this approach is that the mapping between the input Euclidean space and the output feature space realized by neural networks is nonlinear and obscure. Thus it is common to rely on iterative optimization methods with local linearization such as Gauss-Newton [4]. The performance of these approaches deteriorates when the initial pose difference gets larger. A method is called *global* if the registration result is independent of initial pose, or *local* otherwise [5]. Most existing correspondence-free methods are local.

In this paper, we employ equivariant neural networks and implicit shape learning into correspondence-free registration, achieving two major advantages. First, because of the equivari-

5th Conference on Robot Learning (CoRL 2021), London, UK.

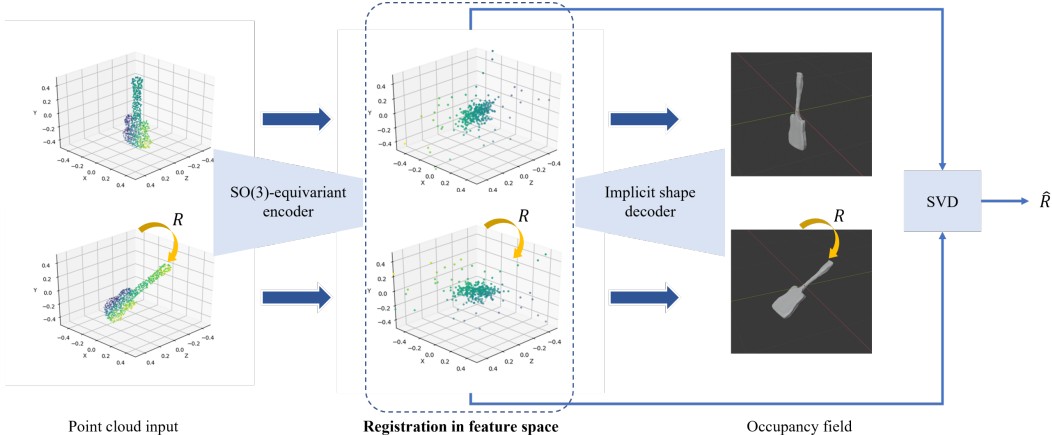

Figure 1: Overview of the correspondence-free rotational registration network. The point cloud input is of shape $\mathbb{R}^{N \times 3}$, and the encoded feature is of shape $\mathbb{R}^{C \times 3}$. $N$ is the number of points, and $C$ is the dimension of features. Occupancy field is a function $v(p) \mapsto [0, 1], p \in \mathbb{R}^3$ mapping any 3D coordinate to an occupancy value (see Sec. 3.2 for details). The rotation is estimated by aligning the features using Horn's method [1].

ance property, the feature space preserves the same rotation operation as the Euclidean input space, allowing us to solve the feature-space registration in a closed form using Horn's method. As a result, our method is *global*, i.e., invariant to the initial pose difference. Second, we improve the registration robustness against noise in point clouds to better handle challenges in real-world applications through training the features for both the implicit shape reconstruction and registration tasks. The implicit shape learning encourages the network to learn features describing the underlying geometry rather than specific points sampled on the geometry; thus, the features are better prepared for registration by including the registration in the training loop. In this work, we focus on the rotational registration task.

In particular, the main contribution of this paper are:

1. We propose a correspondence-free point cloud registration method via SO(3)-equivariance representation learning and closed-form pose estimation [1].

2. The proposed method is global, delivering consistent registration regardless of initial rotation error.

3. The proposed method is robust against imperfections in input point clouds, including noise, sampling, and density differences.

## 2 Preliminaries on Equivariance and Vector Neurons

Symmetry (or equivariance) in a neural network is an important ingredient for efficient learning and generalization. It also allows us to build a clear connection between the changes in the input space and the output space. For example, convolutional neural networks are translation-equivariant. It makes a network generalize to image shifting and enables huge parameter saving. However, other symmetries are less explored. For example, most point cloud processing networks (e.g., Point-Net [6], DGCNN [7]) are not equivariant to rotations.

We leverage the recently proposed Vector Neuron layers [8] to enable rotation-equivariant feature learning for point clouds. For the sake of completeness, we briefly introduce the preliminary on equivariance and the design of layers.

A function $f : X \to X$ is equivariant to a set of transformations $G$, if for any $g \in G$, $f$ and $g$ commutes, i.e., $g \cdot f(x) = f(g \cdot x), \forall x \in X$. For example, applying a translation on a 2D image and then going through a convolution layer is identical to processing the original image with a convolution layer and then shifting the output feature map. Therefore convolution layers are translation-equivariant.

---

[1]Code will be available at `https://github.com/minghanz/EquivReg`.

Vector Neurons extend the equvariance property to SO(3) rotations for point cloud neural networks. The key idea is to augment the scalar feature in each feature dimension to a vector in $\mathbb{R}^3$ and redesign the linear, nonlinear, pooling, and normalization layers accordingly. In a regular network, the feature matrix with feature dimension $C$ corresponding to a set of $N$ points is $V \in \mathbb{R}^{N \times C}$. In Vector Neuron networks, the feature matrix is of form $V \in \mathbb{R}^{N \times C \times 3}$. The mapping between layers can be written as $f : \mathbb{R}^{N \times C_l \times 3} \to \mathbb{R}^{N \times C_{l+1} \times 3}$, where $l$ is the layer index. Following this design, the representation of SO(3) rotations in feature space is straight-forward: $g(R) \cdot V := VR$, where $g(R)$ denotes the rotation operation in the feature space, parameterized by the 3-by-3 rotation matrix $R \in \text{SO}(3)$. In the following, we ignore the first dimension $N$ of $V$ for simplicity.

The linear layer for Vector Neurons is similar to a traditional MLP: $f_{\text{lin}}(V) = WV$, where $W \in \mathbb{R}^{C_{l+1} \times C_l}$. It is easy to see that such a mapping is SO(3)-equivariant:

$$g(R) \cdot f_{\text{lin}}(x) = WVR = f_{\text{lin}}(g(R) \cdot V) \tag{1}$$

Designing nonlinearities for Vector Neuron networks is less trivial. A naive ReLu on all elements in the feature matrix will break the equivariance. Instead, a *vectorized* version of ReLu is designed as the truncation of vectors along a hyperplane. Specifically, given an input feature matrix $V$, predict a canonical direction $k = UV \in \mathbb{R}^{1 \times 3}$ where $U \in \mathbb{R}^{1 \times C}$. Then the ReLU is defined as:

$$v' = \begin{cases} v, & \text{if } \langle v, k \rangle \geq 0 \\ v - \langle v, \frac{k}{\|k\|} \rangle \frac{k}{\|k\|}, & \text{otherwise} \end{cases} \tag{2}$$

for each row $v \in \mathbb{R}^{1 \times 3}$ of $V$. This design preserves SO(3) equivariance because $k$ and $V$ will rotate together, and $\langle VR, kR \rangle = \langle V, k \rangle$, meaning that rotations do not change the activation. For the design of pooling layers and normalization layers of Vector Neurons, please refer to the original paper for details [8].

## 3 Methodology

We employ a SO(3)-equivariant neural network to construct a feature space preserving the same rotation representations as to the input space, thus simplifying solving the rotation for feature alignment. Furthermore, deep implicit representation learning and end-to-end training of registration improve the robustness of the features against noise and allow registration between two different scans of the same geometry.

### 3.1 SO(3)-equivariant point cloud global feature extractor

We choose PointNet as the feature extractor backbone. The permutation-invariant property implies that two point clouds correspond to the same feature embeddings if they only differ in the permutation of points. We replace the layers in PointNet with the corresponding Vector Neuron version [8]. Denote the feature extraction network as $f$, a point cloud $P \in \mathbb{R}^{N \times 3}$, and its rotated and permuted copy $P' = MPR$, where $M \in \mathbb{R}^{N \times N}$ is a permutation matrix and $R \in \text{SO}(3)$ is a 3-by-3 rotation matrix. Then we have

$$Q' = f(P') = f(MPR) = f(MP)R = f(P)R = QR \tag{3}$$

where $Q := f(P) \in \mathbb{R}^{C \times 3}$ is the feature of $P$ extracted by the encoder. Equation (3) is guaranteed by the SO(3)-equivariance and permutation-invariance properties, which is essential for our feature-space registration.

A Vector Neuron network is designed to take inputs of $\mathbb{R}^{N \times C_0 \times 3}$, while the point cloud input is $P \in \mathbb{R}^{N \times 3}$. Simply reshaping $P$ to $\mathbb{R}^{N \times 1 \times 3}$ with $C_0 = 1$ does not work, because a linear layer with weight $W \in \mathbb{R}^{C_1 \times 1}$ will make the resulted feature $WP \in \mathbb{R}^{N \times C_1 \times 3}$ linearly dependent in the third dimension for each point. Following [8], we use one DGCNN edge-convolution [7] layer to initialize the $\mathbb{R}^{N \times C_0 \times 3}$ feature from raw $\mathbb{R}^{N \times 3}$ point cloud using the neighboring points. See [8] for more details. Furthermore, we explore replacing the k-nearest neighbor operation with a random sampling of k points in a ball with a certain radius in the edge convolution layer. The motivation is to alleviate the sensitivity of the feature against point cloud density (see Sec. 4.2.3). However, the evaluations in this paper use k-NN by default unless specified.

## 3.2 Deep implicit representation learning

Being able to register two rotated copies of the same point cloud is not enough for practical use. In real applications, the measurements have noise, and the individual points captured by different scans generally do not correspond to the same physical points.

Our solution is to repurpose the global feature (for registration ultimately) for implicit shape representation. Following the Occupancy Network [9], we build an encoder-decoder network. The encoder is the aforementioned SO(3)-equivariant feature extractor $f$, taking a point cloud $P \in \mathbb{R}^{N \times 3}$ as input. The decoder takes the encoded shape feature $Q = f(P) \in \mathbb{R}^{C \times 3}$ and a queried position $p \in \mathbb{R}^3$ as input, predicting the occupancy value of that position $v(p; Q) \in [0, 1]$. Notice that $p$ need not be coming from $P$. In practice, we sample $n$ querying points for each input to calculate the implicit shape reconstruction (occupancy value prediction) loss:

$$L_{\text{occ}} = \sum_{i=1}^{n} L_{\text{cross\_entropy}}(v(p_i; f(P)), v_{\text{gt}}(p_i)), \tag{4}$$

where $v_{\text{gt}}(p) \in \{0, 1\}$ is the ground truth occupancy value of a query point $p$. The sampling strategy of the querying points is consistent with the Occupancy Network [9]. In this way, we implicitly encourage the feature $f(P_1), f(P_2), ...$ to be close as long as $P_1, P_2, ...$ are point clouds of the same geometry.

## 3.3 Point cloud registration through feature alignment

With the above network design, we can further relax the connection between the two point clouds $P'$ and $P$ in (3). The features $Q$ and $Q'$ are *approximately* rotation-equivariant even when $P$ and $P'$ are not permutations of each other, as long as they are sampled on the same geometry. We say it is approximate because a trained implicit shape reconstruction network can only represent the implicit shape approximately. Therefore, we also include the registration step in the training loop, directly encouraging the accurate registration of point clouds with noise. As shown below, the registration has a closed-form solution and is differentiable, allowing end-to-end training of features.

One can see that (3) is exactly in the form of an orthogonal Procrustes problem. The learned features $Q$ and $Q'$ can be regarded as two *pseudo* point clouds, and each row of them is *automatically matched*. This problem has a standard closed-form solution. First, calculate the cross-covariance matrix $H = Q^{\mathsf{T}}Q'$ followed by its Singular Value Decomposition (SVD) as $H = USV^{\mathsf{T}}$. The optimal rotation matrix $\hat{R}$ is given as

$$\hat{R} = V\Lambda U^{\mathsf{T}}, \tag{5}$$

where $\Lambda = \text{diag}(1, 1, \det(VU^{\mathsf{T}})) \in \mathbb{R}^{3 \times 3}$. We design the registration loss as the mean squared error between the residual rotation matrix and the identity matrix, i.e., chordal distance, as

$$L_{\text{reg}} = \left\| R_{\text{gt}}^{\mathsf{T}} R_{\text{est}} - I_3 \right\|_F^2, \tag{6}$$

where $R_{\text{est}}$ is calculated from (5), and $\|\cdot\|_F$ denotes the Frobenius norm.

# 4 Experiments

## 4.1 Overview

Our network is trained on the ModelNet40 dataset [10]. It is composed of 12311 CAD models from 40 categories of objects. Following the official split, 9843 models are used for training, and 2468 models are used for testing. The models are preprocessed using the tool provided by Stutz and Geiger [11] to obtain watertight and simplified models that are centered and scaled to a unit cube. Given watertight models, the occupancy value of any 3D coordinate can be determined depending on whether it is inside or outside the model.

We also evaluate our method on a real-world indoor dataset: 7Scenes [12], which is a subset of 3DMatch [13]. We choose it to be consistent and comparable with one of our baselines, Feature-Metric Registration (FMR) [14]. 7Scenes consists of 7 sequences of RGBD scans of indoor environments. The RGB information is not used in the experiments.

Table 1: Rotational registration error given rotated copies of point clouds. Tested using ModelNet40 official test set. The best are shown in **bold**. The second best are shown in *italic*. All values are in degrees.

| Max initial rotation angle | | | 0 | 30 | 60 | 90 | 120 | 150 | 180 |
|---|---|---|---|---|---|---|---|---|---|
| Categories | Global | Methods | Rotation error after registration | | | | | | |
| Correspondence-free | N | PCR-Net[15] | 7.08 | 9.50 | 27.38 | 68.22 | 109.61 | 129.29 | 133.49 |
| | N | FMR[14] | **0.00** | 0.45 | 5.29 | 21.95 | 42.26 | 66.39 | 79.43 |
| | Y | Ours | *0.02* | **0.02** | **0.02** | **0.02** | **0.02** | **0.02** | **0.02** |
| Correspondence-based | N | RPM-Net[16] | 0.26 | 0.27 | 0.42 | 1.57 | 2.85 | 3.42 | 4.02 |
| | Y | DeepGMR[5] | *0.02* | **0.02** | **0.02** | **0.02** | **0.02** | **0.02** | **0.02** |

The training of the network is composed of two stages. The network is first trained for occupancy value prediction only, after which the network is trained for the occupancy prediction and registration tasks jointly. In the first stage, 1024 points are randomly sampled on each mesh model as the input point cloud with batch size 24. In the second stage, a pair of point clouds are sampled on the same shape model with a different pose and number of points. The registration loss between the pair is calculated, together with the occupancy loss of each of the point clouds. The occupancy value is queried at 2048 points in the unit cube. The batch size of such pairs is 10 in this stage.

The dimension of the learned global feature is set as $C = 342$, corresponding to $342 \times 3 = 1026$ dimensions for a traditional feature vector, close to the dimension chosen by previous literature including PointNetLK [3], PCRNet [15], and FMR [14]. The network is trained on a single Tesla V100 GPU.

Representative correspondence-free methods PCRNet and FMR are our baselines. PointNetLK also falls in this category, which we omit here because FMR follows a very similar approach to PointNetLK and can be viewed as its improved version. We also included results from some representative correspondence-based methods to give readers a better context of both types of approaches. The experiments on the baselines are based on their official open-source implementation and pretrained weights.

In all following experiments, random initial rotations are generated using the axis-angle representation, of which the angle is sampled from a uniform distribution with the specified maximum, and the rotation axis is also randomly picked. The evaluation metric is the mean isotropic rotation error as in [16].

## 4.2   Evaluation on synthetic data: ModelNet40

### 4.2.1   Registration with rotated copies of point clouds

We first tested the registration of two points clouds that are only different by rotation and permutation. Both point clouds have 1024 points sampled from the mesh model. The results are shown in Table 1. We can see that as local methods, both PCR-Net and FMR only work properly when the initial rotation is small because they rely on iterative refinements. In comparison, our method provides consistent and almost perfect registration independent of the initial rotation angle.

Both correspondence-based methods, RPM-Net and DeepGMR, work well in all initial conditions. DeepGMR delivers similarly perfect accuracy as ours, both of which are global methods. The error of RPM-Net slightly increases as the initial rotation gets larger while remaining small overall, probably because the inputs are noise-free, and perfectly matching pairs of points exist in the data, allowing the correspondence-finding process to work nicely.

### 4.2.2   Registration with rotated copies of point clouds corrupted with Gaussian noise

In practice, the pair of point clouds for registration will not be identical to each other. Therefore we test the case where the source and target point clouds are both corrupted by noise. We add a Gaussian noise with $\sigma = 0.01$ to all points before putting them through the networks. The rest of the setup is the same as in Sec. 4.2.1. As shown in Table 2, PCR-Net and FMR show a similar trend of increasing error when the initial angle gets larger. Our method, though showing a slightly larger registration error, still outperforms the baseline in most columns. Most importantly, the error remains consistent across different initialization.

Table 2: Rotational registration error of noisy point clouds on ModelNet40.

| Max initial rotation angle | | | 0 | 30 | 60 | 90 | 120 | 150 | 180 |
|---|---|---|---|---|---|---|---|---|---|
| Categories | Global | Methods | Rotation error after registration | | | | | | |
| Correspondence-free | N | PCR-Net[15] | 7.21 | 9.30 | 27.40 | 69.96 | 108.43 | 125.55 | 132.40 |
| | N | FMR[14] | 1.56 | 2.01 | 5.78 | 19.95 | 45.73 | 67.65 | 82.05 |
| | Y | Ours | 2.84 | 2.95 | *2.79* | *3.58* | *3.39* | *3.40* | *3.26* |
| Correspondence-based | N | RPM-Net[16] | **0.22** | **0.24** | 3.14 | 23.14 | 52.20 | 84.31 | 109.59 |
| | Y | DeepGMR[5] | *1.73* | *1.74* | **1.76** | **1.77** | **1.75** | **1.74** | **1.74** |

Table 3: Rotational registration error of point clouds with different densities on ModelNet40.

| Max initial rotation angle | | | 0 | 30 | 60 | 90 | 120 | 150 | 180 |
|---|---|---|---|---|---|---|---|---|---|
| Categories | Global | Methods | Rotation error after registration | | | | | | |
| Correspondence-free | N | PCR-Net[15] | 7.37 | 9.70 | 27.04 | 65.95 | 109.22 | 126.43 | 130.37 |
| | N | FMR[14] | *1.95* | *2.71* | *6.91* | 23.75 | 45.14 | 66.97 | 81.56 |
| | Y | Ours | 16.16 | 15.20 | 16.59 | **15.74** | **16.93** | **17.16** | **16.50** |
| Correspondence-based | N | RPM-Net[16] | **0.48** | **0.52** | **3.53** | 23.60 | 52.70 | 84.07 | 109.62 |
| | Y | DeepGMR[5] | 19.60 | 19.50 | 19.67 | *19.98* | *19.33* | *19.24* | *19.74* |

As the point-wise correspondence is corrupted by noise, RPM-Net deteriorates fast as the initial rotation grows. DeepGMR performs the best, showing the benefit of global methods and that its component-wise correspondence is more robust to noise. It also shows a small gap between our correspondence-free method and the state-of-the-art correspondence-based global method, motivating future improvements.

### 4.2.3 Registration with point clouds with different densities

To further test the robustness of the proposed method against variations in the point clouds, we tested the registration performance when given two point clouds of different densities. It is a practical challenge when the sensor changes or one conducts frame-to-model alignments. Here we sample 1024 and 512 points for the pair of point clouds, respectively. The result is shown in Table 3. We do observe an increase of error in our method in this experiment. A possible reason is that the DGCNN layer for feature initialization at the beginning of the encoder is sensitive to the point cloud density since it takes the relative coordinate of neighboring points as inputs. Therefore, we also experimented with replacing the k-nearest neighbor with random sampling in a ball centered at each point when collecting neighboring points for the edge-convolution of that point. A more detailed discussion is in the supplementary material.

With either of the strategies, the registration error is slightly larger than that of the baselines when the initial rotation angle is small but is much smaller when the initial rotation is arbitrary. We notice a similar trend in the correspondence-based methods. RPM-Net outperforms DeepGMR if the initial rotation is small but performs much worse otherwise. Between the two global methods, DeepGMR shows a slightly larger error than ours.

### 4.3 Evaluation on real-world data: 7Scenes

The model evaluated on 7Scenes is first trained on ModelNet40 with the two stages mentioned in Sec. 4.1, and then finetuned on 7Scenes with the registration loss only. The train/test split is consistent with the work of Huang et al. [14].

The experiments on 7Scenes consist of two parts. In the first part, the pair of point clouds are from the same frame but sampled differently. In the second part, the pair of point clouds are from adjacent frames and only overlaps partially. They reflect challenges in real-world applications since any change of the viewpoint of the sensor could result in different samples of points and visible parts. Due to the page limit, we only show the result of the first part here, and the second part is in the supplementary material.

### 4.3.1 Registration with point clouds that are sampled differently

The original dense RGBD scan has more than 100,000 points. 1024 points are sampled randomly to form an input point cloud. The pair of point clouds then go through random rotation to form the final input. In this case, the points in one point cloud are neither a noisy version nor a subset of

Table 4: Rotational registration error of point clouds sampled differently on 7Scenes.

| Max initial rotation angle | 0 | 30 | 60 | 90 | 120 | 150 | 180 |
|---|---|---|---|---|---|---|---|
| Methods | Rotation error after registration | | | | | | |
| FMR[14] | **1.591** | **1.605** | 6.614 | 19.829 | 39.496 | 68.463 | 84.597 |
| Ours | 4.701 | 4.319 | **4.932** | **6.070** | **5.474** | **5.189** | **5.388** |

the other. To our knowledge, we are one of the first to report registration results under this setting in correspondence-free methods [3, 14, 15]. Table 4 shows that our method is working properly on real-world scans of point clouds that are different samplings of the underlying geometry.

Overall, our proposed rotational registration method can estimate consistent results independent of initial rotations. It outperforms the correspondence-free baselines in all evaluation settings when the initial rotational angle is arbitrary. Experiments on synthetic and real-world data are conducted, covering conditions with noisy inputs, varying density, different sampling, and partial overlapping. However, our method still shows some error increase when challenging inputs are presented, motivating future improvements.

## 5 Related Work and Discussion

### 5.1 Correspondence-based point cloud registration

A major challenge here is to recover the corresponding pairs of points from a pair of point clouds. ICP simply matches the closest points together, solves the transformation, and iteratively rematches the closest points after aligning the two point clouds using the estimated transformation [2]. Point-to-line [17], point-to-plane [18], plane-to-plane [19], and Generalized-ICP [20] build local geometric structures to the loss formulation.

Finding correspondences require strong feature descriptor for points, on which deep learning approaches show expertise. Through metric learning, a feature descriptor can be learned such that matching points are close to each other in the feature space, while non-matching points are far away. Approaches following this idea include PPFNet [21], 3DSmoothNet [22], SpinNet [23], and FCGF [24]. Good point matching leads to accurate pose estimation when solving the orthogonal Procrustes problem. Therefore, the error in pose estimation can be used to supervise point matching and feature learning. DCP [25], RPM-Net [16], DGR [26], and 3DRegNet [27] are some of the methods leveraging ground truth pose to supervise point feature learning. Neural networks are designed to better pick the keypoints (e.g., USIP [28] and SampleNet [29]). Keypoint selection and feature learning may also be considered jointly. Representative works include 3DFeat-Net [30], D3Feat [31], DeepICP [32], and PRNet [33].

The main remaining challenge is that such matching pairs may not exist in the input point clouds in the first place, because point clouds are sparse and a point may not be captured repeatedly by different scans. Soft matching (or many to many) [34, 35] is proposed to address this problem, but it is at best an approximation of the underlying true matching using sparse samples, which can deteriorate the performance when the sparsity increases. DeepGMR [5] learns latent Gaussian Mixture Models (GMMs) for point clouds and establishes component-to-component correspondences.

### 5.2 Correspondence-free point cloud registration

Correspondence-free registration treats a point cloud as a whole rather than a collection of points, requiring a global representation for an entire point cloud. In this way, the limitation brought by the sparsity as mentioned in Sec. 5.1 is circumvented. CVO [36, 37] represents a point cloud as a function in a reproducing kernel Hilbert space, transforming the registration problem to maximizing the inner product of two functions. A series of deep-learning-based methods attempt to extract a global feature embedding for a point cloud and solve the registration by aligning the global features. Examples include PointNetLK [3], PCRNet [15], and Feature-Metric Registration [14]. A limitation of global-feature-based registration is that the nonlinearity of the feature extraction networks leaves us few clues about the structure and properties of the feature space. Therefore these methods rely on iterative local optimization such as the Gauss-Newton algorithm. Consequently, these methods require good initialization to achieve decent results. Our method differentiates from previous work

in that the feature extraction network is $SO(3)$ equivariant, enabling well-behaved optimization in the feature space (detailed in Sec. 3).

## 5.3 Group equivariant neural networks

Neural networks today mainly preserve symmetry against translation, limiting the capacity of networks to deal with broader transformations presented in data. One line of work forms kernels as some steerable functions so that the rotations in the input space can be transformed into rotations in the output space [38, 39, 40]. However, this strategy can limit the expressiveness of the feature extractor since the form of the kernels is constrained.

Another strategy is to *lift* the input space to a higher-dimensional space where the group is contained so that equivariance is obtained naturally [41, 42]. However, for each lifted input to $K$ group elements, we need to integrate over the entire group for computing its convolution. While being mathematically sound, this approach involves the use of group $\exp$ and $\log$ maps (for mapping from and to the Lie algebra) and a Monte Carlo approximation to compute the convolution integral over a fixed set of group elements [42, Sections 4.2-4.4].

Cohen and Welling [43] studied the problem of learning the irreducible representations of commutative Lie groups. Kondor and Trivedi [41] studied the generalization of equivariance and convolution in neural networks to the action of compact groups. The latter also discussed group convolutions and their connection with Fourier analysis [44, 45]. In this context, the convolution theorem naturally extends to the group-valued functions and the representation theory [46].

A new design of $SO(3)$ equivariant neural networks is proposed by Deng et al. [8]. This approach allows incorporating the $SO(3)$ equivariance property into existing network architectures such as PointNet and DGCNN by replacing the linear, nonlinear, pooling, and normalization layers with their *vector neuron* version. Our equivariant feature learner builds on this simple and useful idea.

## 5.4 Deep implicit shape modeling

Implicit shape modeling represents a shape as a field on which a function such as an occupancy or signed distance function is defined. The surface is typically defined by the equation $f(x, y, z) = c$ for some constant $c$. In recent years, deep models are developed to model such implicit function fields, where a network is trained to predict the function value given a query input point coordinate. Some representative works include DeepSDF [47], OccNet [9], NeRF [48].

Implicit functions model a shape continuously, up to the infinite resolution, which offers opportunities in registration to circumvent the challenge in data association due to sparsity in point clouds. DI-Fusion [49] leverages deep implicit models to do surface registration and mapping by optimizing the position of the query points to minimize the absolute SDF value. iNeRF [50] performs RGB-only registration through deep implicit models by minimizing the pixel residual between observed images and neural-rendered images. In this work, we also exploit the continuous nature of the deep implicit model but by using the learned latent feature instead of the decoded function field.

## 6 Conclusion

In this paper, we presented a correspondence-free rotational registration method for point clouds. The method is built upon the developments in equivariant neural networks and implicit shape representation learning. We construct a feature space where the rotational relations in Euclidean space are preserved, in which registration can be done efficiently. We circumvent the need for data association and solve the rotation in closed form, achieving low errors independent of initial rotation angles. Furthermore, we leverage implicit representation learning and end-to-end registration to enhance the network robustness against non-ideal cases where the points are not exactly corresponding.

We conducted experiments on synthetic and real-world data, showing the network's ability to handle various input imperfections while also motivating future work for improved performance. Furthermore, there are two open issues unsolved. The first problem is on generalizing the $SO(3)$-equivariant neural network to $SE(3)$ so that the registration of rotation and translation can be solved together. The second problem is on generalizing the registration to handle outdoor LiDAR scans. These problems are left for future studies.

## Acknowledgment

Toyota Research Institute provided funds to support this work. Funding for M. Ghaffari was in part provided by NSF Award No. 2118818.

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
