# OpenReview forum: "Correspondence-Free Point Cloud Registration with SO(3)-Equivariant Implicit Shape Representations"
_robot-learning.org/CoRL/2021/Conference — CoRL2021 Poster_

### Official Review · Reviewer_nLiN · 2021-07-19

**Originality:** Good
**Technical Quality:** Fair
**Clarity Of Presentation:** Very Good
**Impact:** 2

**Recommendation:**

Weak Accept: I recommend accepting the paper, but will not argue for my recommendation if the majority of other reviewers have a different opinion.

**Summary:**

This paper presents a method for registering a pair of point clouds without correspondences (data association). The key idea is to use the so-called vector neurons to embed the original 3D point clouds into a 3D feature space that is equivariant to 3D rotations, and invariant to permutations. Precisely, if the original point cloud is rotated by an arbitrary rotation and permuted by an arbitrary permuation, then the learned features will be rotated by the same rotation, but will not be affected by the permutation. Using such an architecture, estimating the rotation can be performed in feature space using Horn's method in closed form.

To train the features, the paper proposes to use an encoder-decoder architecture with the encoder being the vector neurons and the decoder being an implicit shape representation for computing the losses more robustly. The proposed method is tested on ModelNet40, compared with two baselines, and shown to achieve generally better performance in the presence of noise and density variances.

**Issues:**

See weaknesses.

**Reviewer Expertise:**

Excellent: Expert knowledge on the topic of the paper

**Strengths And Weaknesses:**

Strengths:
- The SO(3)-equivariant network using vector neurons is a very interesting idea. I like this idea because, as written in the summary, it discards the permutation/ordering of the point cloud, but preserves the rotation so that registration can be done in the feature space.  In some sense, I think the network can be seen as simultaneously learning keypoints and correspondences, because the output features of the network, Q and Q' as in eq. (3), can be seen as a set of corresponding keypoints, where the keypoints do not necessarily belong to the original point cloud (kind of being hallucinated by the network). But I am OK with the authors calling this method "correspondence-free".

- This paper is very easy to understand, with very few typos. The preliminaries and methodology sections well explain the key novelty of the paper.

Weaknesses
- The major weakness of this paper is that experiments are too toy and simple to convince me that the proposed method is useful for any practical robotics or computer vision applications. The authors write in conclusions that "generalizing to deal with translation" and "generalizing to large-scale problems" are two directions of future work. However, I cannot agree with accepting a paper which proposes a new method without extensive evaluation on practical problems, and I think those two generalizations must be done before this paper can be accepted.
- To be more specific, firstly, the ModelNet40 dataset is a toy dataset that only contains single objects without complex cluttered scenes typically encountered in real applications. Second, the experiments in this paper do NOT consider point clouds with partial overlapping, which is too ideal to be useful. In fact, Table 3 shows that the proposed method already gives poor performance under different densities. So I have reason to suspect this method will perform even worse under partial overlapping and cluttered scenes. Third, I think it is also important to compare the proposed method to correspondence-based methods.  Indeed, [Ref1] (see Table 1) shows that strong correspondence-based approaches like TEASER [Ref2] already achieves almost 100% recall on ModelNet40 and the problem without partial overlapping is considered more or less a solved problem.  Therefore, I suggest to test on the more realistic 3DMatch dataset [Ref3], with comparison to more baseline approaches to demonstrate the usefulness of the proposed method.
- I think Section 3.2 Deep implicit representation learning should be presented in more details for the reader to fully understand how the loss function is computed.

Technical suggestions
I want to provide a few known ideas that may help the proposed method.
- It is shown in [Ref2] that robust point cloud registration can be solved in a decoupled fashion (scale, rotation and then translation) by designing invariant measurements. Therefore, the idea in [Ref2] already gives one approach for generalizing the proposed method to handle translations.

- As I mentioned, the proposed method can be seen as simultaneously learning keypoints and correspondences. Therefore, using this correspondence-based perspective, it is straightforward to apply robust estimation methods to register the learned features. Precisely, currently the authors use Horn's method to estimate rotation from Q and Q' in eq. (3). However, Horn's method is known to be sensitive to outliers (wrong matches in Q and Q'). So I believe the authors could try using a robust method to estimate R from Q and Q', such as RANSAC (available in Open3D) or TEASER [Ref2], which perhaps can help get more accurate estimates.


[Ref1] Yuan, Wentao, Benjamin Eckart, Kihwan Kim, Varun Jampani, Dieter Fox, and Jan Kautz. "Deepgmr: Learning latent gaussian mixture models for registration." In European Conference on Computer Vision, pp. 733-750. Springer, Cham, 2020.

[Ref2] Yang, Heng, Jingnan Shi, and Luca Carlone. "Teaser: Fast and certifiable point cloud registration." IEEE Transactions on Robotics 37, no. 2 (2020): 314-333.

[Ref3] Zeng, Andy, Shuran Song, Matthias Nießner, Matthew Fisher, Jianxiong Xiao, and Thomas Funkhouser. "3dmatch: Learning local geometric descriptors from rgb-d reconstructions." In Proceedings of the IEEE conference on computer vision and pattern recognition, pp. 1802-1811. 2017.


**Summary Of Recommendation:**

I think this paper proposes an interesting and novel technical idea that may have good impact.

However, the experiments are significantly lacking to convince me the practical value of the proposed method. In its current form, the paper cannot be accepted.

**Post rebuttal**

The authors added new experiments on 7Scenes, which addressed my major concern. Although the experiments are still not particularly strong, considering the idea of this paper is interesting, I'd like to increase my score to Weak accept.

---

> ### Author Response · Authors · 2021-08-31
> **Response to reviewer nLiN**
>
> > Q1: The major weakness of this paper is that experiments are too toy and simple to convince me that the proposed method is useful for any practical robotics or computer vision applications. The authors write in conclusions that "generalizing to deal with translation" and "generalizing to large-scale problems" are two directions of future work. However, I cannot agree with accepting a paper which proposes a new method without extensive evaluation on practical problems, and I think those two generalizations must be done before this paper can be accepted.
>
> A1: Thanks for the comment. We expanded the experiment part with evaluation on the 3DMatch dataset. Specifically, we evaluated on the 7Scenes dataset which is a part of 3DMatch, so as to be consistent with the FMR [1] baseline. We added two settings for the evaluation: 1. Registration between point clouds that are different samples of the geometry, instead of rotated copies, noisy copies, or subset/superset of each other (see Section 4.3.1). 2. Registration between point clouds that only partially overlap (see Section 4.3.2). The result of Section 4.3.1 (Table 5) shows average rotation error of about 4-5 degrees, indicating that our method works well with real-world data, and our method is robust to the different samplings of the point clouds. The result of Section 4.3.2 (Table 6) shows average rotation error of about 24 degrees. While it is a substantial degradation, it outperformed the FMR baseline on all initial rotation conditions. To our knowledge, we are among the first to report quantitative results on real-world point clouds of different samplings and partial overlaps in correspondence-free methods.
> However, since our focus of the paper is on leveraging the rotational equivariance property to do deep feature registration, we limit our discussion to rotational registration. We have an ongoing work extending the SO(3) equivariance property to SE(3), which will then enable full rotation and translation registration.
>
> > Q2: To be more specific, firstly, the ModelNet40 dataset is a toy dataset that only contains single objects without complex cluttered scenes typically encountered in real applications.
>
> A2: As mentioned in A1, we extended experiments to real-world dataset 7Scenes (part of 3DMatch).
>
> > Q3: Second, the experiments in this paper do NOT consider point clouds with partial overlapping, which is too ideal to be useful. In fact, Table 3 shows that the proposed method already gives poor performance under different densities. So I have reason to suspect this method will perform even worse under partial overlapping and cluttered scenes.
>
> A3: As mentioned in A1, we included experiments on partially overlapping point clouds in Section 4.3.2.
>
> > Q4: Third, I think it is also important to compare the proposed method to correspondence-based methods. Indeed, [Ref1] (see Table 1) shows that strong correspondence-based approaches like TEASER [Ref2] already achieves almost 100% recall on ModelNet40 and the problem without partial overlapping is considered more or less a solved problem. Therefore, I suggest to test on the more realistic 3DMatch dataset [Ref3], with comparison to more baseline approaches to demonstrate the usefulness of the proposed method.
>
> A4: As mentioned in A1, we extended experiments to real-world dataset 7Scenes (part of 3DMatch).
>
> >Q5: I think Section 3.2 Deep implicit representation learning should be presented in more details for the reader to fully understand how the loss function is computed.
>
> A5: Thanks for the advice. Now Section 3.2 is extended with more details, and the loss function is introduced in Equation 4.

---

> > ### Comment · Reviewer_nLiN · 2021-09-01
> > **Thanks for the response**
> >
> > Thanks for the response and the added experiments. I have increased my score.

---

### Official Review · Reviewer_ZnCf · 2021-07-22

**Originality:** Good
**Technical Quality:** Excellent
**Clarity Of Presentation:** Excellent
**Impact:** 3

**Recommendation:**

Strong Accept: I recommend accepting the paper and will argue for my recommendation even if other reviewers hold a different opinion.

**Summary:**

This paper describes a correspondence-free method for pointcloud registration.  The main idea is to learn an SO(3)-equivariant (and permutation-invariant) mapping from the (entire) input pointcloud into an abstract feature space.  The fact that the learned feature mapping is SO(3)-equivariant means that the desired rotation R aligning the pointclouds can be gotten as the optimal rotation aligning their *features*; this latter problem admits a simple direct closed-form solution using Horn’s method.

In detail, the paper shows how to “upgrade” the (permutation-invariant) PointNet to be SO(3)-equivariant by replacing its layers with the [SO(3)-equivariant] Vector Neurons proposed in reference [5].  This modified PointNet is then used as the encoder in an encoder-decoder network (based on OccupancyNet) that is designed to learn a (global) implicit shape descriptor for the pointcloud.  The idea behind the use of an implicit shape representation is to account for small nuisance variations in the sampled pointclouds (due to both measurement noise, and the fact that in general two scans of the same object will not sample *exactly* the same sets of points on the object).  The overall result is a network design that learns an implicit shape representation that is SO(3)-equivariant *by construction*.

Experimentally, the paper evaluates the proposed method in comparison with PCR-Net and Feature-Metric Registration (FMR) [two correspondence-free deep pointcloud registration methods] on three tasks requiring the registration of pairs of pointclouds sampled from CAD models taken from the ModelNet40 dataset.  Overall, the results show that (as expected) the accuracy of the registrations computed by the SO(3)-equivariant approach proposed in this work is *independent* of the magnitude of the rotation aligning the input pointclouds.  This is *not* true for PCR-Net and FMR, whose performance significantly degrades with increasing alignment angle because they make use of (local) iterative refinement to align the input pointclouds.  In absolute terms, the proposed method performs very well in the presence of large-magnitude rotations and Gaussian measurement noise, although its performance does seem to degrade significantly when comparing two pointclouds of differing density.

**Issues:**

The only issue that I would specifically encourage the authors to address during the response and revision period is to try to improve the performance of the method on the differing-density case: I think this would greatly strengthen the paper, as well as (more importantly) making the method significantly more useful.

**Reviewer Expertise:**

Good: General knowledge of the area

**Strengths And Weaknesses:**

My overall impression of this work is very positive: the problem under consideration is certainly well-motivated, the proposed approach is very elegant, and the empirical results show that it achieves the desired result [namely, SO(3)-equivariance] -- as in fact it must, since the proposed method inherits this property *by construction*.  I would be very interested in using this method in my own work (which I think is about the highest praise a reader can bestow).

The paper as a whole is also remarkably well-written; I particularly appreciated its clear and concise review of equivariance and Vector Neurons (the latter of which I had not seen before) and the thorough review and contextualization of related work in Section 5.

In my view the paper’s only substantial weakness (which it already acknowledges) is the significant degradation in performance in aligning pointclouds of different sampling densities.  I do not completely agree with the paper’s assertion (in lines 179-180) that pointcloud registration errors of ~20 degrees are “practical” -- I can easily imagine applications (e.g. object manipulation or localization) where such errors would be too large for the returned result to be practically useful.  [I would also expect this different-density scenario to arise fairly regularly, since it simply corresponds to scanning the same object from two different distances.]  Nevertheless, I still think that the SO(3)-equivariant design reported in this work is a very solid technical contribution that is likely to be of interest to the robotics community, despite this specific shortcoming.

A couple of other minor comments:

* I notice that in the results reported in Table 5.3, FMR seems to perform very well as long as the initial rotational offset is less than ~30 degrees.  Given that the proposed method is capable of generating registrations accurate to within about 20-25 degrees across *all* initial offsets, perhaps one way of improving robustness in the differing-densities case would be to treat the proposed method as an *initialization*, and then apply a subsequent (finer) refinement using FMR or something similar?  (Admittedly this lacks the elegance of having a single end-to-end SO(3)-equivariant method, but as a practical approach it might be very effective at addressing the (uniformly) higher errors reported in the differing-density case.)

* In lines 104-106, the paper mentions that this method is among the first to address the problems of measurement noise and inexact correspondences between the input pointclouds in deep correspondence-free registration methods.  If this is correct, I would encourage the authors to emphasize this point more strongly (e.g. in the abstract and introduction), as it strikes me as an additional significant feature of the method.



**Summary Of Recommendation:**

My recommendation for accepting this paper is based upon the following considerations:

* It addresses a fundamental problem in robotic perception (pointcloud alignment), which is likely to be of interest to a very broad subset of the community
* It develops a solution to this problem that provides a novel set of capabilities (DNN-based correspondence-free poincloud registration that is *also* SO(3)-equivariant) with respect to prior work
* The design process that leads to the proposed method is both mathematically rigorous and technically very interesting, and is based on principles [e.g. geometric analysis of the design of feature maps, equivariance, etc.] that can be generalized far beyond the specific problem and algorithm considered in this work.  I expect that many readers would find this independently interesting, and I could easily imagine that the presentation of these ideas in a nice "case study" like this paper could catalyze their application to many other problems.

Finally, I want to mention that I rated this paper's impact at "3" only because the criteria for 4 strikes me as unreasonably high (even the fraction of *published* papers that have "a major impact in robotics or machine learning" is vanishingly small -- well under 5%), and there was no intermediate rating between "incremental" and "major".  This should in no way be construed as a criticism of the paper's merit -- it's a very fine contribution, and I would like to see it accepted.

---

> ### Author Response · Authors · 2021-08-31
> **Response to reviewer ZnCf**
>
> > Q1: In my view the paper’s only substantial weakness (which it already acknowledges) is the significant degradation in performance in aligning pointclouds of different sampling densities. I do not completely agree with the paper’s assertion (in lines 179-180) that pointcloud registration errors of ~20 degrees are “practical” -- I can easily imagine applications (e.g. object manipulation or localization) where such errors would be too large for the returned result to be practically useful. [I would also expect this different-density scenario to arise fairly regularly, since it simply corresponds to scanning the same object from two different distances.]
>
> A1: Thanks for pointing it out. We further address this issue by including the registration in the training loop and supervise the network with registration loss (see Section 3.3). The error under different density is now reduced to around 15 degrees (see Table 3). By exploring different neighboring point sampling strategies in the feature initialization layer (see Section 4.2.3), the registration error under different density could be further reduced to around 13 degrees (see Table 4).
>
> > Q2: I notice that in the results reported in Table 5.3, FMR seems to perform very well as long as the initial rotational offset is less than ~30 degrees. Given that the proposed method is capable of generating registrations accurate to within about 20-25 degrees across all initial offsets, perhaps one way of improving robustness in the differing-densities case would be to treat the proposed method as an initialization, and then apply a subsequent (finer) refinement using FMR or something similar? (Admittedly this lacks the elegance of having a single end-to-end SO(3)-equivariant method, but as a practical approach it might be very effective at addressing the (uniformly) higher errors reported in the differing-density case.)
>
> A2: Thanks for the suggestion. One solution we tried is to adopt the iterative optimization algorithms (e.g. Gauss-Newton as in PointNetLK, FMR) as a post-processing step. However, naively adding this step does not work because SVD already gives the globally optimal solution in the sense of feature mean square error theoretically. GN may better handle outliers by using a robust norm like Huber’s norm as the error term, but we did not find it improving our results in the experiments. Therefore we do not include this part of discussion in the paper. Another way is as you mentioned, appending a whole FMR network after initialization using our method. I included it in the discussion at the end of Section 4.2.3, but did not report experiments on it, since it is kind of not closely related to our key contributions. Still, thanks for the advice!
>
>
> > Q3: In lines 104-106, the paper mentions that this method is among the first to address the problems of measurement noise and inexact correspondences between the input pointclouds in deep correspondence-free registration methods. If this is correct, I would encourage the authors to emphasize this point more strongly (e.g. in the abstract and introduction), as it strikes me as an additional significant feature of the method.
>
> A3: Thank you for pointing it out. Now the experiments are extended with testing under differently sampled point clouds and partially overlapped ones, therefore should better support the claim. However, the claim is now moved to Section 4.3.1 and 4.3.2 due to reorganization.

---

### Official Review · Reviewer_sqvt · 2021-07-24

**Originality:** Fair
**Technical Quality:** Good
**Clarity Of Presentation:** Poor
**Impact:** 3

**Recommendation:**

Weak Reject: I recommend rejecting the paper, but will not argue for my recommendation if the majority of other reviewers have a different opinion.

**Summary:**

This paper proposes to do point cloud registration in learned feature space. The features are extracted using a neural network that retains SO(3)-equivariance in points' features. Then the registration is done in the feature space and then propagated to every point. Experiments are done on the point clouds of the synthetic dataset ModelNet40.

**Issues:**

The authors can address the items under "Weakness" and "Some other issues" in my main review section.

**Reviewer Expertise:**

Very good: Comprehensive knowledge of the area

**Strengths And Weaknesses:**

Strength:

- The idea of doing point cloud registration in feature space is novel. This could be inspiring for the rest of the research community.
- The experiment results seem to be strong and outperform the baselines by a significant margin.

Weakness:
- The extracted deep features could be very unstable, especially when there is occlusion in the point cloud.
- The authors only evaluated two simple real-data effects: random noise in the point cloud and different densities. However, these two cases are both way too simple and easy, because if the point occupancy grid is used as representation, both cases do not change occupancy much. One very important experiment is missing: registration of $\textit{partial}$ 3D point scans, especially when the two scans have different partial pieces. This is a case where the proposed algorithm can fail completely because partial scans can dramatically change the deep features, therefore the registration step in the feature space can fail easily. The authors have to evaluate this case to support their claim in the paper.
- Since the authors used occupancy grid as point representation, the proposed method is unable to deal with large-scale point scans such as depth scans of the entire room, because the grid resolution will be severely limited.
- What is the loss function used to train the deep features? This is not clear from the paper.

Some other issues on the paper:
- Fig 2. visualization of equivariant features is unnecessary.
- The related work section is too long. Most of the sentences in this section can be more concise.
- "Occupancy Network" is first mentioned in line 108, but is not cited until line 141. This is very confusing.
- In Fig 1., the author mentioned the output to be "Occupancy Field", but where is it defined in the text?
- How is "Rotation Error" defined in Tables 1, 2, and 3? Is it rotation angle difference, in terms of radians or degrees? If so, why isn't rotation axis difference evaluated? Or it is some combination of rotation axis and angle? It's not clear from the paper.



**Summary Of Recommendation:**

I'm not convinced that the proposed approach is actually useful, especially because the authors used point occupancy as representation. This results in the proposed approach being unable to deal with large-scale scans.

Besides, the presentation of this paper is very poor. A lot of necessary details are missing from the paper and are very hard to follow. It seems that this paper is written in a rush without careful check of logic and soundness.

---

> ### Author Response · Authors · 2021-08-31
> **Response to reviewer sqvt**
>
> > Q1: The extracted deep features could be very unstable, especially when there is occlusion in the point cloud.
>
> A1: Thanks for pointing it out. We understand that the rotational equivariance property does not guarantee good features for alignments when the inputs could be corrupted in various ways in real-world scenarios. We address this issue by including the registration inside the training loop (see Section 3.3), and use imperfect point clouds as inputs. Through supervision of the registration loss, the network can learn more robust features when the inputs are corrupted. We use the partially overlapping point cloud pairs in 7Scenes dataset (part of 3DMatch) as inputs to finetune the network for evaluation on 7Scenes. While the partial-overlapping inputs still result in worse registration accuracy, it outperforms the baseline under all initial rotations (see Table 6).
>
> We understand that correspondence-free methods need to come up with a better way to deal with occlusions/partial overlaps if they rely on some global feature for registration. Recent developments in correspondence-based methods (PREDATOR [1]) predict the overlapping part of a pair of point clouds. This idea could also benefit correspondence-free methods, by encouraging the network to extract features from the overlapping part only. However, this is out of the scope of this paper and will be saved for future work.
>
> > Q2: The authors only evaluated two simple real-data effects: random noise in the point cloud and different densities. However, these two cases are both way too simple and easy, because if the point occupancy grid is used as representation, both cases do not change occupancy much. One very important experiment is missing: registration of 3D point scans, especially when the two scans have different partial pieces. This is a case where the proposed algorithm can fail completely because partial scans can dramatically change the deep features, therefore the registration step in the feature space can fail easily. The authors have to evaluate this case to support their claim in the paper.
>
> A2: We extended the experiment part to evaluate on the 7Scenes (part of 3DMatch) dataset, consisting of RGBD scans of indoor environments.  The tests are under two settings: 1. Registration between point clouds that are different samples of the geometry, instead of rotated copies, noisy copies, or subset/superset of each other (see Section 4.3.1). 2. Registration between point clouds that only partially overlap (see Section 4.3.2). The result of Section 4.3.1 (Table 5) shows average rotation error of about 4-5 degrees, indicating that our method works well with real-world data, and our method is robust to the different samplings of the point clouds. The result of Section 4.3.2 (Table 6) shows average rotation error of about 24 degrees. While it is a substantial degradation, it outperformed the FMR baseline on all initial rotation conditions. To our knowledge, we are among the first to report quantitative results on real-world point clouds of different samplings and partial overlaps in correspondence-free methods.
>
>
> > Q3: Since the authors used occupancy grid as point representation, the proposed method is unable to deal with large-scale point scans such as depth scans of the entire room, because the grid resolution will be severely limited.
>
> A3: Thanks for proposing the concern. Our approach is actually not using any discretization or grid map. The occupancy network learns a function mapping arbitrary querying 3D coordinates to a scalar in [0, 1] representing the occupancy value. During training, the query points are sampled randomly. Therefore, our method does not have scalability issues regarding any potential grid map. Besides, the network’s model size is also independent of the number of points in the input point cloud.
>
> > Q4: What is the loss function used to train the deep features? This is not clear from the paper.
>
> A4: Thanks for pointing it out. We currently use the occupancy prediction loss and registration loss in the training. They are now shown in the paper as Equation 4 and 6. More specifically, we train the network in two stages. In the first stage we only train with the occupancy prediction loss. In the second stage, both losses are used, and the features are expected to be better suited for registration while maintaining its ability for occupancy prediction. See details in Section 4.1 starting at Line 155.

---

> > ### Author Response · Authors · 2021-08-31
> > **Response to reviewer sqvt 2**
> >
> > > Q5: Fig 2. visualization of equivariant features is unnecessary.
> >
> > A5: Thanks for the suggestion. Fig 2 is now removed, as it seems not to be a very integrated part of the paper.
> >
> > > Q6: The related work section is too long. Most of the sentences in this section can be more concise.
> >
> > A6: Thanks for the comment. The related work section (Section 5) is now shortened and hopefully more concise.
> >
> > > Q7: "Occupancy Network" is first mentioned in line 108, but is not cited until line 141. This is very confusing.
> >
> > A7: Thanks for pointing it out. The citation is now added (now at Line 116).
> >
> > > Q8: In Figure 1, the author mentioned the output to be "Occupancy Field", but where is it defined in the text?
> >
> > A8: Thanks for pointing it out. “Occupancy Field” is now explained in the caption of Figure 1 and further explained in Section 3.2.
> >
> > > Q9: How is "Rotation Error" defined in Tables 1, 2, and 3? Is it rotation angle difference, in terms of radians or degrees? If so, why isn't rotation axis difference evaluated? Or it is some combination of rotation axis and angle? It's not clear from the paper.
> >
> > A9: The rotation error is quantified by the angle of the residual rotation between the ground truth rotation and the estimated rotation, in the axis-angle representation. We added it at Line 170. Such representation only considers the magnitude of the residual rotation. Rotation axis difference between the GT and estimation is handled in this error term, because wrongly estimated rotation axis will result in large residual rotation, therefore large angle in axis-angle representation. The error is in degree, as shown in the caption of Table 1.

---

### Official Review · Reviewer_b5A4 · 2021-07-24

**Originality:** Good
**Technical Quality:** Fair
**Clarity Of Presentation:** Good
**Impact:** 3

**Recommendation:**

Weak Reject: I recommend rejecting the paper, but will not argue for my recommendation if the majority of other reviewers have a different opinion.

**Summary:**

The paper proposes a correspondence-free Deep Neural Network (DNN)-based method for point cloud registration. The network exploits the "Vector Neurons" architecture, a SO(3)-equivariant DNN for point cloud processing, to extract a global feature vector from an input point cloud. A decoder network is used to reconstruct the original input from the global feature and a query position. Finally, the rotation between two point clouds is estimated by comparing their global feature vectors in a closed-form solution.

**Issues:**

- A strong evidence, such as a comparison of the proposed approach with recent correspondence-based methods, or at least a strong argument on why correspondence-free methods should be preferred over correspondence-based ones, should be provided.
- An experimental evaluation on a real-world dataset, such as 3DMatch or KITTI/KITTI-360, would give a better insight into the possible applications of the approach.

**Reviewer Expertise:**

Very good: Comprehensive knowledge of the area

**Strengths And Weaknesses:**

Strengths:
+ The paper is well written, well organized, and in general easy to read.
+ The proposed approach is very interesting, and addresses a relevant problem.
+ Different methods for point cloud registration are well presented and discussed.
+ The results show great improvement over existing correspondence-free DNN-based approaches, especially when the initial rotation between the point clouds is large.

Weaknesses:
- The paper does not provide a strong argument or evidence in support of using correspondence-free over correspondence-based methods. The argument that "matching (point) pairs may not exist in the input point clouds" (line 210) is weak, since robust estimators such as RANSAC can effectively overcome this issue. Recent correspondence-based point cloud registration methods [1, 2, 3] that combine DNN-based feature extractors with RANSAC achieved outstanding results and robustness to initial misalignment. A strong evidence, such as a comparison of the proposed approach with recent correspondence-based methods, or at least a strong argument on why correspondence-free methods should be preferred over correspondence-based ones, should be provided.
- The proposed method only estimates the rotation between the two point clouds, while most existing approaches estimate both rotation and translation.
- The proposed method is evaluated only on the synthetic ModelNet dataset, an experimental evaluation on a real-world dataset, such as 3DMatch or KITTI/KITTI-360, would give a better insight into the possible applications of the approach.

[1] Bai, X., Luo, Z., Zhou, L., Fu, H., Quan, L., & Tai, C. L. (2020). D3feat: Joint learning of dense detection and description of 3d local features.

[2] Cattaneo, D., Vaghi, M., & Valada, A. (2021). LCDNet: Deep Loop Closure Detection and Point Cloud Registration for LiDAR SLAM.

[3] Huang, S., Gojcic, Z., Usvyatsov, M., Wieser, A., & Schindler, K. (2021). PREDATOR: Registration of 3D Point Clouds with Low Overlap.

**Summary Of Recommendation:**

Although the proposed method is interesting, a strong evidence supporting the claim that correspondence-free methods should be preferred over correspondence-based ones is missing. Therefore, my recommendation is weak reject.

---

> ### Author Response · Authors · 2021-08-31
> **Response to reviewer b5A4**
>
> > Q1: The paper does not provide a strong argument or evidence in support of using correspondence-free over correspondence-based methods. The argument that "matching (point) pairs may not exist in the input point clouds" (line 210) is weak, since robust estimators such as RANSAC can effectively overcome this issue. Recent correspondence-based point cloud registration methods [1, 2, 3] that combine DNN-based feature extractors with RANSAC achieved outstanding results and robustness to initial misalignment. A strong evidence, such as a comparison of the proposed approach with recent correspondence-based methods, or at least a strong argument on why correspondence-free methods should be preferred over correspondence-based ones, should be provided.
>
> A1: Thanks for the comment. We do not claim that our method should be preferred over correspondence-based ones, or that correspondence-free methods should be preferred over correspondence-based ones. Today’s correspondence-based methods are developed very well and have good performance in real world applications. In comparison, correspondence-free methods are less mature and generally work in simpler conditions. However, it is always attractive to look for “simpler” solutions. Attempting to remove the middle step of data association is consistent with this trend. Therefore, this work is trying to push forward the status of correspondence-free methods, as an alternative way of solving the registration problem.
>
> > Q2: The proposed method only estimates the rotation between the two point clouds, while most existing approaches estimate both rotation and translation.
>
> A2: Thanks for the comment. We limit our discussion to rotational registration because our focus is on leveraging the property of rotational equivariance of the specific type of network in registration. While it may not prepare itself ready for practical robotic applications, we think it is worth sharing the idea to inspire further developments in this direction. We simultaneously have an undergoing research trying to extend the SO(3) equivariant network to SE(3) equivariance, which will hopefully enable full 6-dimensional registration.
>
> > Q3: The proposed method is evaluated only on the synthetic ModelNet dataset, an experimental evaluation on a real-world dataset, such as 3DMatch or KITTI/KITTI-360, would give a better insight into the possible applications of the approach.
>
> A3: Thanks for the advice. We agree that experiments on real data could better show the practical value of this approach. We extended the experiment section (Section 4) with registration results on the 3DMatch dataset. Following FMR, we use the 7Scenes dataset, which is part of the 3DMatch dataset, recording RGBD sequences of indoor enrironments. We added two settings for the evaluation: 1. Registration between point clouds that are different samples of the geometry, instead of rotated copies, noisy copies, or subset/superset of each other (see Section 4.3.1). 2. Registration between point clouds that only partially overlap (see Section 4.3.2). The result of Section 4.3.1 (Table 5) shows average rotation error of about 4-5 degrees, indicating that our method works well with real-world data, and our method is robust to the different samplings of the point clouds. The result of Section 4.3.2 (Table 6) shows average rotation error of about 24 degrees. While it is a substantial degradation, it outperformed the FMR baseline on all initial rotation conditions. To our knowledge, we are among the first to report quantitative results on real-world point clouds of different samplings and partial overlaps in correspondence-free methods.

---

> > ### Comment · Reviewer_b5A4 · 2021-09-02
> > **Response to authors**
> >
> > Thanks for the response. I think that the added experiments on the 7Scenes dataset are a great addition to the paper.
> >
> > I think that your response on correspondence-free and correspondence-based methods should be included in the paper, and ideally a comparison against a correspondence-based method should be included. This will highlights how much your approach improved existing correspondence-free methods, and how far correspondence-free methods are still from correspondence-based ones.
> >
> > The inclusion of a proper discussion on correspondence-free vs. correspondence-based methods (ideally with evaluations), and the already included experiments on a real-world dataset, would make the paper good for publication.

---

> > > ### Author Response · Authors · 2021-09-02
> > > **Further response to reviewer b5A4**
> > >
> > > Thanks for the comment. We will include our response on correspondence-free and correspondence-based methods and experiment results on correspondence-based methods in the final version of the paper if accepted or in the future version of the work in general.

---

### Official Review · Reviewer_4hsK · 2021-08-01

**Originality:** Very Good
**Technical Quality:** Good
**Clarity Of Presentation:** Good
**Impact:** 4

**Recommendation:**

Weak Accept: I recommend accepting the paper, but will not argue for my recommendation if the majority of other reviewers have a different opinion.

**Summary:**

This paper proposes a new encoder-decoder approach for correspondence-free point cloud registration with rotations. The encoder is a combination of PointNets which provides permutation invariance and the recently proposed Vector Neurons which provides SO3 equivariance. The decoder is an implicit function (occupancy network) that takes an encoding and a query position and predicts the occupancy value. Two point clouds are registered by treating their encodings as a pair of pseudo point clouds that are already pairwise matched, so that the registration problem admits a closed form solution.  Compared to two other baselines, when the initial rotational difference is large, the proposed method shows significantly lower error.

**Issues:**

- Figure 1 and its caption could be improved to be more informative.
- Overall, some technical details on architecture design are missing section 3.
- Modern autodiff engines allow Eqn 4 to be implemented and treated as a "layer" in a network. It was not clear whether the feature alignment step is part of the training or outside it?
- In Table 2 and Table 3, the error seems higher than the baselines for small initial rotation angles. How can this be fixed?

**Reviewer Expertise:**

Good: General knowledge of the area

**Strengths And Weaknesses:**

Strengths

- The paper is generally well written and proposes a novel point cloud rotational registration method.
- The improvements in rotational error seem quite significant particularly for large initial rotation angles.

Weaknesses

- Many details on network architecture and training are omitted and delegated to the appendix, making some sections a bit hard to follow.
- In Table 2 and Table 3, the error seems higher than the baselines for small initial rotation angles. How can this be fixed?

**Summary Of Recommendation:**

Point cloud registration has many applications in Robotics. Though a robotics application is not shown in this paper, the network design is particularly interesting in the way it addresses permutation and rotational invariance, and builds on implicit shape representations. The improvements seem convincing overall.

---

> ### Author Response · Authors · 2021-08-31
> **Respond to reviewer 4hsK**
>
> > Q1: Figure 1 and its caption could be improved to be more informative.
>
> A1: Thanks for the advice. Figure 1 is modified to include the registration block. Its caption is extended to describe the overall structure of the method, giving a more concrete explanation on the input and output of each stage.
>
> > Q2: Overall, some technical details on architecture design are missing section 3. Many details on network architecture and training are omitted and delegated to the appendix, making some sections a bit hard to follow.
>
> A2: Thank you for the comment. Section 3 is expanded with more details of the network architecture and the loss functions used. The I/O connections among different components are stated using consistent notations of the variables. Introduction of the training strategy is expanded in Section 4 (experiments).
>
> > Q3: Modern autodiff engines allow Eqn 4 to be implemented and treated as a "layer" in a network. It was not clear whether the feature alignment step is part of the training or outside it?
>
> A3: It is a great question. At the time of the first submission, the feature registration step is not part of the training loop. Inspired by your question, we experimented including the registration of a pair of differently sampled point clouds of the same object in the training. This change improved the registration accuracy when noise is presented and when the density of the pair of point clouds is different. More details are given in Section 3.3 and the new results can be found in Table 2 and 3. Thanks for the comment!
>
> > Q4: In Table 2 and Table 3, the error seems higher than the baselines for small initial rotation angles. How can this be fixed?
>
> A4: By including the registration step in the training loop, the error of our method is reduced in Table 2 and 3, but is still slightly higher than the baselines for small initial rotation angles. One solution we came up with is to adopt the iterative optimization algorithms (e.g. Gauss-Newton as in PointNetLK, FMR) as a post-processing step. However, naively adding this step does not work because SVD already gives the globally optimal solution in the sense of feature mean square error theoretically. GN may better handle outliers by using a robust norm like Huber’s norm as the error term, but we did not find it improving our results in the experiments. Therefore we do not include this part of discussion in the paper.
>
> It is possible that a non-equivariant network as in PointNetLK or FMR is more flexible and can be better fitted in a small range of transformation perturbations to better handle a noisy pair of point clouds with small initial rotation angles. One may use our method as initialization and then go through a PointNetLK or FMR network for refinement. It is briefly mentioned at Line 206 in Section 4.2.
> In terms of the performance drop under different densities, we think that it could be related to the edge convolution in the feature initialization layer in the encoder (see Line 102 at Section 3.1 for more details). We explored replacing the k-NN sampling with random sampling of nearby points inside a local ball to alleviate the sensitivity against density change (see Section 4.2.3 for more details). It successfully lowered the registration error under density differences, but also increased the error when the point clouds are rotated copies (see Table 4). The reason could be that the random sampling introduced randomness in the feature initialization layer, slightly breaking the equivariance property. We have not found a perfect method to reduce the registration error under all input conditions. An ideal feature initialization layer should be deterministic and insensitive to the density, and we will keep exploring for a better answer.

---

### Author Response · Authors · 2021-08-31
**Response to meta review**

Thanks for your time organizing the review. We appreciate that the novelty of the proposed method is recognized. The mentioned weaknesses and issues are very insightful, and we benefited a lot from them. By addressing the concerns, we improved the methodology, experiments, and writing of the paper. The advice by reviewers helped improve our result, or provided inspiration for future directions.
Overall, the modification we made in response to the reviews are of three folds:

More experiments, especially on real-world dataset (7Scenes of 3DMatch, see Section 4.3), and with more realistic settings (different samplings, and partial overlap).

Improved method. We now include the registration in the training loop (Section 3.3), which helped improve the registration accuracy compared with the initial version (Table 2 and 3). We also explored a new feature initialization strategy (Section 4.2.3) to improve the robustness against density changes (Table 4).

More details in the paper. Details on the network architecture, loss functions, and training strategy are presented in Section 3 and 4.

>Q1: For the paper to have an impact on the robotics community, the experiments should include evaluations on real-world datasets (such as 3DMatch and KITTI), and evaluate on partial/occluded point clouds. It is also recommended to include comparisons against correspondence-based methods (LCDNet, PREDATOR, TEASER or similar).

A1: Thanks for the advice. We expanded the experiment part with evaluation on the 3DMatch dataset. Specifically, we evaluated on the 7Scenes dataset which is a part of 3DMatch, so as to be consistent with the FMR [1] baseline. We added two settings for the evaluation: 1. Registration between point clouds that are different samples of the geometry, instead of rotated copies, noisy copies, or subset/superset of each other (see Section 4.3.1). 2. Registration between point clouds that only partially overlap (see Section 4.3.2). The result of Section 4.3.1 (Table 5) shows average rotation error of about 4-5 degrees, indicating that our method works well with real-world data, and our method is robust to the different samplings of the point clouds. The result of Section 4.3.2 (Table 6) shows average rotation error of about 24 degrees. While it is a substantial degradation, it outperformed the FMR baseline on all initial rotation conditions. To our knowledge, we are among the first to report quantitative results on real-world point clouds of different samplings and partial overlaps in correspondence-free methods.
We admit that the result may lag behind state-of-the-art correspondence-based methods on real-world data. However, its improvement in the category of correspondence-free methods is substantial, and we believe that it is still an unanswered question whether data association is an inevitable middle step for high-quality registration. Therefore, the exploration of correspondence-free methods still has its value although it has not beaten the SOTA correspondence-based methods yet.
Besides, thanks to the comment by reviewers, we found the developments in correspondence-based methods potentially helpful for correspondence-free ones. For example, PREDATOR [2] predicting the overlapping parts of point clouds could help correspondence-free methods extract more related features for partially overlapping inputs. We mentioned it in the result discussion at Line 233 in Section 4.3.2, and left it as a future direction.

---

### Author Response · Authors · 2021-08-31
**Response to meta review 2**

> Q2: The approach seems to have slightly higher errors (compared to baselines) when the initial rotation angles are small. While this is not necessarily a problem, a discussion on why this is the case and how to resolve this issue is desirable. Similarly, the performance of the approach quickly degrades with the sampling density, which may limit real-world applicability.

A2: Thanks for the comment. The initial error is larger than the baselines mainly when the input density varies. It may be due to the feature initialization layer of the encoder, which involves edge convolution of points with their neighbors. When point density changes, the features from neighboring points could change significantly (more details of the feature initialization layer see Line 102 in Section 3.1). To address this issue, we proposed to replace the k-nearest neighbor with random sampling in a local ball when collecting neighboring points. It successfully lowered the registration error under density differences, but also increased the error when the point clouds are rotated copies (see Table 4). The reason is that the random sampling introduced randomness in the feature initialization layer, slightly breaking the equivariance property. See Section 4.2.3 for more details. We have not found a perfect method to reduce the registration error under all input conditions. An ideal feature initialization layer should be deterministic and insensitive to the density, and we will keep exploring for a better answer.
In some other input conditions (e.g., noisy, different sampling), the FMR baseline also has a slight margin over our method when the initial error is very small. The issue is alleviated by including the registration (Horn’s method) inside the training loop (the error numbers in Table 2 and 3 are smaller than those in the first version), but not completely resolved. A possible reason for this is that the non-equivariant networks are more flexible and could be better fitted in a small range of transformation perturbations. It may hint at a possible future direction as to combine the geometric structure of equivariant networks and flexibility of non-equivariant networks for registration.

> Q3: The approach may not scale to large scenes due to the discretization and the size of the grid map. A discussion about scalability is in order.

A3: Thanks for proposing the concern. Our approach is actually not using any discretization or grid map. The occupancy network learns a function mapping arbitrary querying 3D coordinates to a scalar in [0, 1] representing the occupancy value. During training, the query points are sampled randomly. Therefore, our method does not have scalability issues regarding any potential grid map. When the scene gets larger and more complicated, it may require a more efficient sampling strategy to capture a larger occupancy field with fewer query points, but on the other hand, our final target is registration instead of learning the perfect occupancy.
Besides, the network’s model size is independent of the number of points in the input point cloud.

> Q4: The discussion is hard to follow in some points (e.g., details on the architecture design in Section 3). This point should be easy to address.

A4: Thanks for pointing it out. Section 3 is expanded with more details on the network architecture, I/O specification among the different blocks, and the design of loss functions. Section 4.1 is extended with more details on the training strategy. Fig. 1 and its caption is modified to be more informative and better at summarizing the overall structure of the method.

[1] Huang, Xiaoshui, Guofeng Mei, and Jian Zhang. "Feature-metric registration: A fast semi-supervised approach for robust point cloud registration without correspondences." Proceedings of the IEEE/CVF Conference on Computer Vision and Pattern Recognition. 2020.
[2] Huang, Shengyu, et al. "PREDATOR: Registration of 3D Point Clouds with Low Overlap." Proceedings of the IEEE/CVF Conference on Computer Vision and Pattern Recognition. 2021.

---

### Meta-Review · Area_Chair_VBHK · 2021-08-13

**Recommendation:** Accept (Poster)
**Confidence:** 5

**Metareview:**

The paper has obtained 5 expert reviews, which is above average for CORL and will hopefully provide useful inputs to the authors.

The reviewers appreciated the following aspects: The paper is well written, proposes a novel and elegant registration method, and the results confirm such a method has lower errors under large rotations. The reviewers particularly appreciated the idea of mapping the problem into a rotation-equivariant and permutation-invariant feature space where registration becomes straightforward.

The main weaknesses are:
- For the paper to have an impact on the robotics community, the experiments should include evaluations on real-world datasets (such as 3DMatch and KITTI), and evaluate on partial/occluded point clouds. It is also recommended to include comparisons against correspondence-based methods (LCDNet, PREDATOR, TEASER or similar).
- The approach seems to have slightly higher errors (compared to baselines) when the initial rotation angles are small. While this is not necessarily a problem, a discussion on why this is the case and how to resolve this issue is desirable. Similarly, the performance of the approach quickly degrades with the sampling density, which may limit real-world applicability.
- The approach may not scale to large scenes due to the discretization and the size of the grid map. A discussion about scalability is in order.
- The discussion is hard to follow in some points (e.g., details on the architecture design in Section 3). This point should be easy to address.

*Post-rebuttal comments*:
The reviewers and area chair have carefully read the revised paper and rebuttal and there has been an interesting discussion about the merits of the paper. The reviewers mostly agree that the paper has an interesting contribution and advances the literature on deep-learning-based correspondence-free methods. The reviewers and AC also appreciated the additional experiments: they definitely strengthened the paper. On the downside, one reviewer is still concerned that the 7scenes evaluation is not fully satisfactory since it might still hide issues in the registration of point clouds with low overlap. While I share the same concern, I think no paper is perfect and I consider this paper interesting for the CoRL audience. A second issue that I consider very important is to provide a more transparent contextualization with respect to correspondence-based methods: while I agree that learning-based correspondence-free methods are less mature, it is important for the paper to acknowledge that, and to provide suitable pointers to the non-expert reader. Similarly, I would have liked to see a comparison against non-learning-based correspondence-free methods, such as ICP and variants. Since these are relatively minor changes, I trust the authors will add a few sentences to clarify these points in the final submission, therefore I vote to accept this paper.

Minor comments:
- In the literature review, including ICP in the "correspondence-based" methods feels wrong. Correspondence-based methods take the correspondences as input or establish correspondences using descriptors. It should be included in the correspondence-free methods instead.
- Please rephrase the sentence “To our knowledge, we are one of the first to report registration results under this setting in correspondence-free methods” to mention “learning-based correspondence-free methods” (going back to the previous point, ICP has been extensively used for correspondence-free registration).

---

### Decision · Program_Chairs · 2021-09-13

**Decision:**

Accept (Poster)

**Comment:**

The paper has obtained 5 expert reviews, which is above average for CORL and will hopefully provide useful inputs to the authors.

The reviewers appreciated the following aspects: The paper is well written, proposes a novel and elegant registration method, and the results confirm such a method has lower errors under large rotations. The reviewers particularly appreciated the idea of mapping the problem into a rotation-equivariant and permutation-invariant feature space where registration becomes straightforward.

The main weaknesses are:
- For the paper to have an impact on the robotics community, the experiments should include evaluations on real-world datasets (such as 3DMatch and KITTI), and evaluate on partial/occluded point clouds. It is also recommended to include comparisons against correspondence-based methods (LCDNet, PREDATOR, TEASER or similar).
- The approach seems to have slightly higher errors (compared to baselines) when the initial rotation angles are small. While this is not necessarily a problem, a discussion on why this is the case and how to resolve this issue is desirable. Similarly, the performance of the approach quickly degrades with the sampling density, which may limit real-world applicability.
- The approach may not scale to large scenes due to the discretization and the size of the grid map. A discussion about scalability is in order.
- The discussion is hard to follow in some points (e.g., details on the architecture design in Section 3). This point should be easy to address.

*Post-rebuttal comments*:
The reviewers and area chair have carefully read the revised paper and rebuttal and there has been an interesting discussion about the merits of the paper. The reviewers mostly agree that the paper has an interesting contribution and advances the literature on deep-learning-based correspondence-free methods. The reviewers and AC also appreciated the additional experiments: they definitely strengthened the paper. On the downside, one reviewer is still concerned that the 7scenes evaluation is not fully satisfactory since it might still hide issues in the registration of point clouds with low overlap. While I share the same concern, I think no paper is perfect and I consider this paper interesting for the CoRL audience. A second issue that I consider very important is to provide a more transparent contextualization with respect to correspondence-based methods: while I agree that learning-based correspondence-free methods are less mature, it is important for the paper to acknowledge that, and to provide suitable pointers to the non-expert reader. Similarly, I would have liked to see a comparison against non-learning-based correspondence-free methods, such as ICP and variants. Since these are relatively minor changes, I trust the authors will add a few sentences to clarify these points in the final submission, therefore I vote to accept this paper.

Minor comments:
- In the literature review, including ICP in the "correspondence-based" methods feels wrong. Correspondence-based methods take the correspondences as input or establish correspondences using descriptors. It should be included in the correspondence-free methods instead.
- Please rephrase the sentence “To our knowledge, we are one of the first to report registration results under this setting in correspondence-free methods” to mention “learning-based correspondence-free methods” (going back to the previous point, ICP has been extensively used for correspondence-free registration).